# Binge drinking in young people: protocol for a systematic review of neuropsychological, neurophysiological and neuroimaging studies

Briana Lees,[1] Louise Mewton,[1] Lexine Stapinski,[1] Lindsay M Squeglia,[2] Caroline Rae,[3] Maree Teesson[1]

[1]Centre of Research Excellence in Mental Health and Substance Use, National Drug and Alcohol Research Centre, University of New South Wales, Sydney, Australia
[2]Department of Psychiatry and Behavioral Sciences, Medical University of South Carolina, Charleston, South Carolina, United States
[3]Neuroscience Research Australia, University of New South Wales, Sydney, Australia

**Correspondence to**
Briana Lees;
b.lees@unsw.edu.au

## ABSTRACT

**Introduction** Binge drinking is the most common pattern of alcohol use among young people in Western countries. Adolescence and young adulthood is a vulnerable developmental period and binge drinking during this time has a higher potential for neurotoxicity and interference with ongoing neural and cognitive development. The purpose of this systematic review will be to assess and integrate evidence of the impact of binge drinking on cognition, brain structure and function in youth aged 10–24 years. Cross-sectional studies will synthesise the aberrations associated with binge drinking, while longitudinal studies will distinguish the cognitive and neural antecedents from the cognitive and neural effects that are a consequence of binge drinking.

**Methods and analysis** A total of five peer-reviewed databases (PubMed, EMBASE, Medline, PsychINFO, ProQuest) will be systematically searched and the search period will include all studies published prior to 1 April 2018. The search terms will be a combination of MeSH keywords that are based on previous relevant reviews. Study selection will follow the Preferred Reporting Items for Systematic Reviews and Meta-Analyses guidelines and study quality will be assessed using The Grades of Recommendation, Assessment, Development and Evaluation approach. All studies will be screened against eligibility criteria designed to synthesise studies that examined a young binge drinking sample and used neuropsychological, neurophysiological or neuroimaging assessment techniques. Studies will be excluded if participants were significantly involved in other substances or if they had been clinically diagnosed with an alcohol use disorder, or any psychiatric, neurological or pharmacological condition. If available data permits, a meta-analysis will be conducted.

**Ethics and dissemination** Formal ethics approval is not required as primary data will not be collected. The results will be disseminated through a peer-reviewed publication, conference presentations and social media.

**Trial registration number** International Prospective Register for Systematic Reviews (PROSPERO) number: CRD42018086856.

## INTRODUCTION

Alcohol misuse among young people is widely recognised as a global health priority[1] and

### Strengths and limitations of this study

► This systematic review and meta-analysis will be the first to synthesise neuropsychological, neurophysiological and neuroimaging studies examining the developmental impact of binge drinking on cognition, brain structure and function in youth.
► This review will report on cross-sectional and longitudinal data to first identify the cognitive and neural aberrations associated with binge drinking and second to distinguish the antecedents of binge drinking from the effects that may be caused by binge drinking.
► Identified cognitive and neural precursors and consequences of binge drinking will be informative for prevention, early intervention and treatment efforts.
► While studies will be excluded if participants had been clinically diagnosed with an alcohol use disorder, mild alcohol dependence that has not been formally diagnosed may be more prevalent in adolescent binge drinkers, and this may increase the risk of bias in the review towards a binge population that was seeking help or treatment.

has raised concern about the neurotoxic effects of alcohol use on a large scale.[2] Binge drinking is a pattern of alcohol use that brings blood alcohol concentration levels to 0.08 g/dL which typically occurs after the consumption of four or more alcoholic drinks per drinking occasion (ie, at the same time or within a couple of hours of each other) for females and five or more drinks per occasion for males.[3 4] This episodic pattern of drinking, where an individual drinks less frequently but in larger amounts, is most common among adolescents in Western countries.[5–7] For instance, approximately 10%–16% (USA), 23% (UK) and 15% (Australia) of young adults aged 15–17 years report binge drinking in the previous month.[8–10] The prevalence of binge drinking sharply increases from adolescence to young adulthood, with

40%–50% of young adults reporting binge drinking at least monthly.[6 9 11–13] Extreme binge drinking (defined as 10 or more drinks per occasion) is also common, with 29% of young adults in Australia and 16% of US adolescents engaging in this behaviour.[9 14 15] This is concerning because early alcohol use and binge drinking is associated with a myriad of short-term and long-term negative consequences including blackouts, hangovers and alcohol poisoning,[16 17] alcohol and drug use disorders,[18–20] other mental health problems,[21] risky sexual behaviours,[22 23] injuries[24 25] and increased risk of being a victim of assault or accidental death.[26 27]

Studies consistently indicate that alcohol use and misuse during adolescence (10–19 years) and young adulthood (20–24 years) has a higher potential for neurotoxicity and interference with ongoing neural and cognitive development than during later adulthood.[16 25 28–36] This is because adolescence and young adulthood is a vulnerable developmental period characterised by significant neural changes. Although brain size is thought to stabilise around the age of 5 years,[37] important morphometric restructuring and functional neuromaturation continues to occur during adolescence and young adulthood with substantial myelinisation, synaptic refinement and changes in grey and white matter volume until the age of 25 years.[38] The reward and mesolimbic systems mature during mid-adolescence, prior to the development of prefrontal and cognitive control regions which continue to develop into late adolescence.[39–42] This has a twofold effect; first this hypersensitivity to reward during adolescence results in an increased propensity to engage in risky and sensation-seeking activities, including drug and alcohol use.[43] Second, risky drinking during prefrontal brain development may interfere with neuromaturation and translate to ongoing neural aberrations and top-down cognitive processing deficits, reducing youth's ability to enable self-control and resist temptations (inhibition); to see reason, problem solve and consider alternatives (working memory) and to plan and change perspective (cognitive flexibility).[44] Collectively, these changes in cognitive processing may lead to increased motivation to consume alcohol and a decreased ability to regulate this motivation and drinking behaviour. As many of these developmental changes occur in brain regions that appear to be particularly sensitive to alcohol,[45 46] it is critical that research examines the associated negative consequences of risky episodic drinking during a vulnerable developmental period as the neural insults may have ongoing cognitive and behavioural impacts.

The growing concern about alcohol use among young people has led to a significant increase in the number of studies using neuropsychological, neurophysiological and neuroimaging techniques to determine the effects on brain and cognitive development. Over the past decade, there has also been a rise in the number of longitudinal designs that assess young people before they initiate alcohol use and continue to assess them over time as a portion begin to initiate use. These prospective longitudinal studies have made it possible to disentangle the antecedents and consequences of alcohol use in young people. A recent review of longitudinal studies that concentrated on alcohol initiation in adolescence found that reduced grey matter volume (frontal), less white matter volume (cerebellar, nucleus accumbens, anterior cingulate), poor white matter integrity (fronto-limbic), decreased activation during inhibition and working memory tasks and increased reward response (frontal) were antecedents of alcohol use initiation in adolescence.[47] Accelerated decreases in grey matter (frontal, temporal), attenuated white matter development (pons, corpus callosum), poor white mater integrity and increased brain activation during inhibition and working memory tasks were reported consequences following alcohol use initiation. In terms of cognitive domains, poorer inhibitory functioning and working memory were antecedents of alcohol initiation in adolescence while poorer verbal learning and memory, visuospatial functioning, psychomotor speed and working memory were reported effects following alcohol use initiation in adolescence. By distinguishing the antecedents from the consequences of alcohol use initiation, this review provides researchers with specific neural and cognitive domains to target in prevention and treatment efforts.

Considering binge drinking is the dominant pattern of use among young people, it is important to understand the neural and cognitive impact this pattern of drinking has on the developing brain. Several narrative reviews (summarising neuropsychological, neurophysiological and neuroimaging studies[7 17 28 45 48 49]) and one systematic review from 2014 (including only neuroimaging studies[50]) have summarised the recent binge-drinking literature. The systematic review concluded that there were a number of structural changes associated with binge drinking, including smaller grey and white matter volume compared with non-binge drinkers[29 51] and lower white matter integrity across more than 18 white matter regions.[52 53] Functional differences reported in binge drinkers included less activation during a spatial working memory task[54] and abnormal activation during verbal encoding[55] and decision-making tasks.[56] In terms of neuropsychological studies, narrative reviews have concluded that binge drinking is associated with several cognitive deficits, including impairments in verbal, non-verbal and spatial working memory, as well as attention and executive function.[33–35 57] A review of neurophysiological studies found that young binge drinkers displayed latency differences in several event-related potential (ERP) components, including P1, N1, P3, P3b and P450, in response to a number of cognitive functioning tasks.[28] The early positive and negative voltage deflections (P1, N1) reflect initial sensory differences between binge drinkers and non-binge drinkers, while the later components (P3, P3b, P450) reflect differences in the way participants processed the cognitive tasks. Overall, there is a growing evidence base that consistently demonstrates neural and cognitive aberrations associated with binge drinking.

More recently, there has been an increase in the number of prospective longitudinal studies examining the effect of binge drinking among young people. Integrating these new findings is essential to understanding whether the neural and cognitive precursors and consequences of binge drinking are similar or divergent to the domains related to alcohol initiation in adolescence.

The aim of this systematic review is to therefore provide an update on the expanding literature and synthesise the neuropsychological, neurophysiological and neuroimaging literature on binge drinking and neurodevelopment. This review will also address limitations identified in the previous systematic review. The authors of the 2014 systematic review limited their search to one peer-reviewed database, included adolescents aged 10–19 years and included concurrent substance use. Searching a broader range of peer-reviewed databases may identify studies which were potentially missed in the previous review. Expanding the age range to include young people aged 10–24 years aligns with evidence that neuromaturation continues into the mid-20s,[38] as well as the WHO's definition of young people.[58] To examine the specific effect of binge drinking on brain development and functioning, studies should exclude individuals with concurrent regular use of drugs other than alcohol[45 47] as well as exclude samples that characterised drinking based on non-binge, heavy drinking or diagnostic criteria, including alcohol abuse, dependence and alcohol use disorder, due to the heterogeneity of drinking behaviours and related harms.[49] Alcohol use disorder is characterised by continued alcohol use despite clinically significant social and physiological consequences, including substance abuse, affective symptoms and other psychopathology.[59] Therefore, the type, extent and magnitude of the neural and cognitive aberrations associated with alcohol use disorder are likely to differ from those associated with an adolescent, socially functioning binge drinking population. Additionally, no review has systematically integrated neuropsychological, neurophysiological and neuroimaging data. Integrating neuropsychological and neurophysiological studies

with neuroimaging research is crucial because cognitive processes make an important contribution to excessive alcohol consumption[57] and assessing this data conjointly will provide a broader understanding of the impact binge drinking has on brain development and behaviour. Finally, previous reviews have not critically appraised the within-study risk of bias or overall quality of the body of literature.

This review will involve conducting a systematic literature search of cross-sectional and longitudinal studies to assess and integrate the evidence regarding the impact of binge drinking on cognition, brain structure and function, utilising neuropsychological, neurophysiological (electroencephalography (EEG), ERP) and neuroimaging (MRI, functional MRI (fMRI), diffusion tensor images (DTI), magnetic resonance spectroscopy (MRS)) studies. If available data permits, a meta-analysis will be conducted to determine the overall effects of binge drinking on the outcomes of interest. By including cross-sectional studies, we aim to synthesise the cognitive and neural aberrations associated with binge drinking in young people. On the other hand, longitudinal studies that track individuals over time will distinguish cognitive and neural antecedents that predict later binge drinking from the cognitive and neural effects that are a consequence of binge drinking during adolescence and young adulthood (see figure 1 for logic model). Due to the limited number of published longitudinal studies at the time of the previous systematic review, this systematic review will be the first to infer causality. The predisposing and consequential factors may not be mutually exclusive and some of the vulnerability factors that predict binge drinking behaviour may also be further impacted by the initiation and continuation of binge drinking. Importantly, identified precursors of binge drinking will be informative for prevention and early intervention efforts. Meanwhile, by identifying consequences of binge drinking, treatment efforts will be able to pursue targeted cognitive and physiological training to determine whether these neural insults have ongoing cognitive and behavioural impacts or whether they can recover following a decrease in alcohol use.

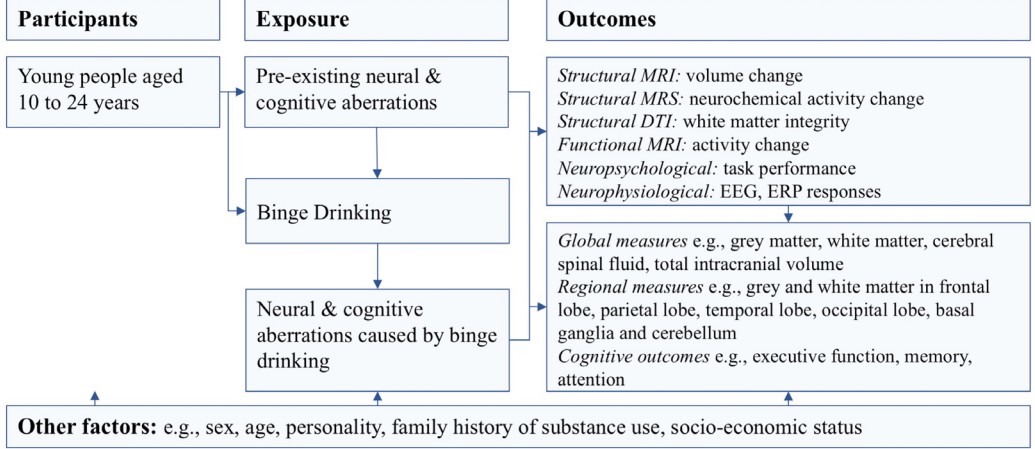

**Figure 1** Logic model: antecedents and consequences of binge drinking in adolescents and young adulthood.

## OBJECTIVES

1. To assess and integrate evidence of the impact of binge drinking on cognitive, structural and functional development in people aged 10–24 years, compared with healthy controls who do not meet the criteria for binge drinking.
2. To synthesise the cognitive and neural aberrations associated with binge drinking by utilising cross-sectional data.
3. To identify the cognitive, structural and functional features that predispose youth to binge drinking and separate this from the cognitive, structural and functional features that may be a consequence of binge drinking.
4. To examine the within-study risk of bias and assess the quality of the body of evidence examining the relationship between binge drinking and cognitive, structural and functional deficits in adolescents and young adults.

### METHODS

This protocol follows the Preferred Reporting Items for Systematic Reviews and Meta-Analysis Protocols (PRISMA-P) statement[60] found in the online Supplementary File. This protocol has been registered with the PROSPERO International Prospective Register of Systematic Reviews of the University of York (registration number: CRD42018086856).

### Search strategy

Relevant literature from PubMed, EMBASE, Medline, PsycINFO and ProQuest will be systematically searched using a comprehensive search strategy which was developed using medical subject headings (MeSH). The Ovid Medline search strategy is provided in the online Supplementary File, which will be replicated for the other electronic databases. This strategy will search through all relevant literature published from database inception to 1 April 2018. A snowballing technique will be employed where the reference list of identified articles will also be screened for suitable studies.

Search terms will be used to identify neuropsychological, neurophysiological and neuroimaging studies assessing the impact of binge drinking on neurodevelopment and neuropsychological task performance in adolescents and young adults. The search terms will be based on previous reviews examining the association between substance use, cognition, brain structure and function.[16 50 61 62] Search terms will be combinations of MeSH keywords describing the participants (adolescent, teenager, youth, emerging adult, young adult), the exposure variable (alcohol, binge drinking, ethanol) and the assessment methods measuring the outcomes of interest (neuroimaging, brain imaging, MRI, fMRI, DTI, MRS, neurophysiological, EEG, ERP, neuropsychological, cognitive, verbal working memory tests, episodic memory tests, visuospatial working memory tests, verbal fluency tests, executive function tests, digit

symbol substitution tests, reaction time, attention). Two reviewers will be involved in independently screening articles, extracting data and assessing the methodological quality.

### Eligibility criteria

Eligibility criteria for this review are defined using population, intervention/exposure, comparator, outcome, and study characteristics. Box 1 provides an overview of the selection criteria.

#### Population

Study samples will be limited by age to human adolescents and young adults ranging from 10–24 years at first assessment, which is consistent with the WHO's definition of young people.[58] Studies that include both a sample of young people and an adult sample (>24 years) will be included if the majority of participants are aged 10–24 years or if a separate analysis for participants within the age range of this review was provided. A minimum of 12 participants per group (binge, comparator) must be included within the study analysis, consistent with a previous review in this area.[50] Finally, studies must be available in the English language to be included in this review.

---

> **Box 1    Selection criteria**
>
> Inclusion criteria
> Population
> 1. Participants aged 10–24 years at first assessment.
> 2. Study is available in the English language.
> 3. n>12 participants per group.
> 4. Human participants (no animal studies).
> Exposure
> 5. Inclusion of binge drinking sample.
> Comparator
> 6. Inclusion of a control group who do not meet criteria for binge drinking.
> Outcomes
> 7. Use of neuropsychological, neurophysiological, structural or functional imaging techniques.
> 8. Presentation of main effects.
> Study characteristics
> 9. Peer-reviewed study.
> 10. Cross-sectional or longitudinal data.
> 11. Empirical data.
> 12. Published before 1 April 2018.
>
> Exclusion criteria
> Exposure
> 13. Studies that involved participants who met criteria for alcohol use disorder.
> 14. Studies that involved participants who were significantly involved with substances other than alcohol.
> 15. Studies that involved participants who had other clinically diagnosed psychiatric, neurological or pharmacological conditions.
> Study characteristics
> 16. Reviews, information in books or letters.

## Exposure

Studies must include a binge drinking sample, where binge drinking is defined as four or more drinks per occasion for females or five or more drinks per occasion for males.[3 4] Consistent with the previous reviews in this area, studies will not be included in this review if samples have ever met diagnostic criteria (eg, alcohol abuse or alcohol use disorder),[49] or if majority of the participants were significantly involved with substances other than alcohol (ie, >5 cannabis use per month, >25 lifetime other drug use occasions[54 63 64]). It is noted that mild alcohol dependence that has not been clinically diagnosed may be more prevalent in adolescent or young adult binge drinkers, and this may result in an increased risk of bias in the review towards a binge-drinking population that was seeking help or treatment compared with binge drinkers who were not. Studies that include participants who smoke tobacco will be included. Participant disclosure of other substance use or a urine sample identifying other substance use will be sufficient to exclude these studies. Studies that included participants who had other clinically diagnosed psychiatric, neurological or pharmacological conditions will also be excluded from this review to ensure that outcomes are specific to binge drinking.

## Comparator

Inclusion of a control group is required for studies to be included in this review. Studies must compare participants who meet the criteria for binge drinking with healthy controls who have never consumed alcohol or who have consumed low levels of alcohol but have never met the criteria for binge drinking.

## Outcomes

Studies must report empirical data where the primary outcomes of interest are global and regional volume (structural images), global and regional activity (functional images; cerebral blood flow or blood oxygen level dependent signal), white matter integrity (DTI), neurochemical activity (MRS; glutamate, gamma-aminobutyric acid, N-acetylaspartate), brain electrical activity (EEG, ERP responses) and cognitive task performance. Global measures include grey matter, white matter, cerebral spinal fluid and total intracranial volume differences between the active and control group. Regional measures include white matter and grey matter (frontal lobe, parietal lobe, temporal lobe, occipital lobe, basal ganglia and cerebellum) differences between the active and control group. For neuroimaging and neurophysiological studies, detailed results of significant findings will be reported. For neuropsychological studies, significant differences in cognitive task performance between the active and control group will be reported.

## Study characteristics

Peer-reviewed cross-sectional and longitudinal neuropsychological, neurophysiological and neuroimaging studies that provide original data and were published before 1 April 2018 will be included. Reviews and information in books or letters will not be included. Any publication that reported data using two or more techniques from the same subject (eg, structural MRI and functional MRI) will be considered separately in the review.

## Selection procedure

Two researchers will be involved in the review and selection procedure. Reviewer one (BL) will screen all titles and abstracts from the peer-reviewed databases to determine eligibility for inclusion in the review. Reviewer two (LM) will independently screen a random selection of 25% of abstracts to ensure accuracy in the study selection. Cohen's kappa will be calculated to assess the interrater agreement between the two reviewers. To ensure a strong level of agreement, a Cohen's kappa of at least 0.8 is required.[65] Full-text versions of the potentially eligible studies will be assessed by both reviewers to further determine eligibility for inclusion. Again, Cohen's kappa will be calculated at the full-text screening stage. Consultation between reviewers will be held at the time of abstract screening and full-text assessment to reconcile any differences of opinion. If consensus cannot be reached, a third member of the research team (LS) will review the eligibility of the study.

## Data extraction

All citations will be imported into Covidence[66] and Endnote.[67] Endnote will be used to store and manage all review data. Covidence will be used to screen titles, abstracts and full texts. Reviewer one will extract data using a data extraction spreadsheet in Excel. Study characteristics will be extracted from published papers, with study authors contacted in the event of missing data. The following information will be extracted from the included studies.

1. *Study information:* names of authors, year of publication, primary outcome measurements, statistical approaches.
2. *Participant characteristics:* sample size, sex, age, handedness, other substance use (ie, tobacco use, cannabis use that is <5 occasions/month).
3. *Alcohol characteristics:* age of onset, frequency of binge drinking, mean quantity of alcohol consumed.
4. *Study characteristics:* imaging modality and analysis, binge drinking sample and control group criteria, cognitive task performed, cognitive and neural domain measured, neurophysiological activity measured, rest/active condition (for functional imaging studies) and exclusion criteria, including the absence of neurological, psychiatric or pharmacological conditions, alcohol use disorder or significant involvement with substances other than alcohol.
5. *Results:* results of outcomes of interest for this review.

## Data analysis and quality assessment

A table summarising the results will be produced, including information about imaging modality and

analysis or neuropsychological tests, sample information, alcohol characteristics, and the study findings. For longitudinal studies, pre-existing cognitive, structural and functional features will be separated from cognitive, structural and functional features that are evident as a consequence of binge drinking. If available data permits, a meta-analysis will be conducted using comprehensive meta-analysis. Hedges' g will be calculated to determine the binge drinking between-group standardised mean effect size from outcomes of interest (global and regional volume, white matter integrity, neural activity and cognitive performance). A random-effects model will be adopted as wide variations in participant characteristics and methodological factors are expected between the studies.

In the case of insufficient homogenous data, a narrative synthesis of the main results extracted from the studies will be completed. The studies will be classified according to the study type (ie, neuropsychological task, neurophysiological measurement, structural imaging, functional imaging) and a summary of differences identified in the binge drinking sample compared with the control group will be reported in text.

Following data extraction, the quality of each study will be critically appraised using The Grades of Recommendation, Assessment, Development, and Evaluation (GRADE) approach.[68] The GRADE system entails an assessment of the quality of a body of evidence for each individual outcome. The GRADE approach defines the quality of the body evidence as the extent to which one can be confident that an estimate of effect or association is close to the quantity of specific interest. This involves considerations of within-study risk of bias (methodological quality of design), directness of evidence, heterogeneity of results, precision of results and the probability of publication bias. Reviewer one will critically appraise all included studies using the GRADE system. Reviewer two will assess the quality of a random selection of 25% of studies to ensure scoring accuracy. Consultation between reviewers will be held to reconcile any differences of opinion.

## Patient and public involvement

Patients and the public were not involved in this systematic review protocol.

## Ethics and dissemination

Ethical approval is not required for this study. The systematic review will be published in a peer-reviewed journal, presented at conferences and will be shared on social media platforms.

## CONCLUSION

This paper summarises the protocol for a systematic review of neuropsychological, neurophysiological and neuroimaging studies conducted in youth who binge drink. The purpose of this review is to assess and integrate the evidence of the developmental impact of binge drinking on cognition, brain structure, and function. Cross-sectional studies will be included in order to synthesise the cognitive and neural aberrations associated with binge drinking in young people. Longitudinal data will be sought to distinguish cognitive and neural antecedents of binge drinking from the cognitive and neural effects that are a consequence of binge drinking during adolescence and young adulthood. This review will be the first to synthesise neuropsychological, neurophysiological and neuroimaging evidence in a systematic way, to include a meta-analysis of the findings, and the first to assess the quality of the body of neuropsychological, neurophysiological and neuroimaging studies. This review aims to provide researchers, policy makers and programme developers with identified antecedents and consequences of binge drinking that will be informative for prevention, early intervention and treatment efforts.

**Contributors** BL conceptualised the study and is the guarantor of the review. BL, LM and LS developed the study design and protocol. LMS, CR and MT provided feedback on the study design and protocol. BL wrote the first draft of the manuscript. All authors read, revised and approved the final manuscript.

**Funding** BL is funded by the University of New South Wales and received a University of New South Wales Scientia PhD Scholarship. This research received no specific grant from any funding agency in the public, commercial or not-for-profit sectors.

**Competing interests** None declared.

**Patient consent** Not required.

**Provenance and peer review** Not commissioned; externally peer reviewed.

**Data sharing statement** This paper does not include original data.

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
