## [Reviewer comments · BMJ Open]

ARTICLE DETAILS

TITLE (PROVISIONAL)	Binge drinking in young people: Protocol for a systematic review of neuropsychological, neurophysiological and neuroimaging studies
AUTHORS	Lees, Briana; Mewton, Louise; Stapinski, Lexine; Squeglia, Lindsay; Rae, Caroline; Teesson, Maree

VERSION 1 – REVIEW

REVIEWER	Daniel F. Hermens Daniel Hermens PhD, Professor of Youth Mental Health & Neurobiology, Sunshine Coast Mind and Neuroscience - Thompson Institute UNIVERSITY OF THE SUNSHINE COAST, 12 Innovation Parkway, Birtinya, Queensland 4575 Australia
REVIEW RETURNED	04-May-2018

GENERAL COMMENTS	This is a very well written paper with a very sound methodology for the proposed systematic (and possibly meta-analytic) review on binge drinking in young people. This paper and the subsequent review will make excellent contributions to the literature. The authors should be commended on the quality and rigour of their approach to this review. I have one recommendation: the authors may consider to include magnetic resonance spectroscopy studies in the proposed review? I found one minor typographical error: In the 'Objectives' - "3. To identify the cognitive, structural, and functional features that predispose youth to binge drinking, and separate this from the cognitive, structural and functional features that are a consequences of binge drinking". this should be "... that are consequences of..." or "... that are a consequence of...". However, a more conservative (less causal) statement would be: "... that may be a consequence of...". Overall, I found this to be an informative, well structured paper and I look forward to seeing the subsequent review being published in due course.
--

REVIEWER	Carsten Grimm Bradford Districts CCG, UK, University of Bradford, UK
REVIEW RETURNED	05-Jun-2018

GENERAL COMMENTS	I am not convinced that the rationale for excluding AUD (alcohol use disorder) is sufficiently explained, especially as in this study individuals might have fulfilled the criteria during adolescence and young adulthood, but have since had normalised drinking patterns or
--

	abstinence. The grey zone of harmful drinking patterns/mild alcohol dependence is possibly more prevalent in the study population, but not formally diagnosed, therefore there is a risk of bias towards a population that was seeking help and treatment at the time vs those who have not.
--	---

VERSION 1 – AUTHOR RESPONSE

Reviewer #1:

1. The authors may consider to include magnetic resonance spectroscopy studies in the proposed review.

We have now included magnetic resonance spectroscopy studies into the inclusion criteria for this systematic review, see page 7: "This review will involve conducting a systematic literature search of cross-sectional and longitudinal studies to assess and integrate the evidence regarding the impact of binge drinking on cognition, brain structure, and function, utilising neuropsychological, neurophysiological (electroencephalography; EEG, event-related potential; ERP) and neuroimaging (magnetic resonance imaging; MRI, functional MRI; fMRI, diffusion tensor images; DTI, magnetic resonance spectroscopy; MRS) studies".

2. I found one minor typographical error: In the 'Objectives' - "3. To identify the cognitive, structural, and functional features that predispose youth to binge drinking, and separate this from the cognitive, structural and functional features that are a consequences of binge drinking". this should be "... that are consequences of..." or "... that are a consequence of...". However, a more conservative (less causal) statement would be: "... that may be a consequence of...".

This typographical error has been fixed: "3. To identify the cognitive, structural, and functional features that predispose youth to binge drinking, and separate this from the cognitive, structural and functional features that may a consequences of binge drinking".

Reviewer #2:

1. I am not convinced that the rationale for excluding AUD (alcohol use disorder) is sufficiently explained, especially as in this study individuals might have fulfilled the criteria during adolescence and young adulthood, but have since had normalised drinking patterns or abstinence. The grey zone of harmful drinking patterns/mild alcohol dependence is possibly more prevalent in the study population, but not formally diagnosed, therefore there is a risk of bias towards a population that was seeking help and treatment at the time vs those who have not.

We have updated the rationale for excluding AUD from the review, on page 6-7: "To examine the specific effect of binge drinking on brain development and functioning, studies should ... exclude samples that characterised drinking based on non-binge, heavy-drinking or diagnostic criteria, including alcohol abuse, dependence and alcohol use disorder, due to the heterogeneity of drinking behaviours and related harms (1). Alcohol use disorder is characterised by continued alcohol use despite clinically significant social and physiological consequences, including substance abuse, affective symptoms and other psychopathology. Therefore, the type, extent and magnitude of the neural and cognitive aberrations associated with alcohol use disorder are likely to differ from those associated with an adolescent, socially functioning binge drinking population."

We agree with Reviewer #2 that there may be an exaggerated grey zone where there may be study participants who meet criteria for mild alcohol dependence but have not been formally diagnosed. To

minimise the possibility of including participants who had met diagnostic criteria at any point prior to the study, we have updated the exposure and comparator criterion:

Page 9 - exposure: "Studies will not be included in this review if samples have ever met diagnostic criteria (e.g., alcohol abuse or alcohol use disorder)"

Page 10 - comparator: "Studies must compare participants who meet the criteria for binge drinking with healthy controls who have never consumed alcohol or who have consumed low levels of alcohol but have never met the criteria for binge drinking".

Furthermore, we have now identified this as a limitation of the review in the 'Strengths and limitations of this study' section and on page 9-10.

Strengths and limitations: "Mild alcohol dependence that has not been formally diagnosed may be more prevalent

in adolescent binge drinkers, and this may increase the risk of bias in the review towards a binge population that was seeking help or treatment".

Page 9-10: "It is noted that mild alcohol dependence that has not been clinically diagnosed may be more prevalent in adolescent or young adult binge drinkers, and this may result in an increased risk of bias in the review towards a binge drinking population that was seeking help or treatment compared to binge drinkers who were not".

- (1) Cservenka A, Brumback T. The Burden of Binge and Heavy Drinking on the Brain: Effects on Adolescent and Young Adult Neural Structure and Function. *Frontiers in Psychology*. 2017;8:1111.